# Rheological Properties of MWCNT-Doped Titanium-Oxo-Alkoxide Gel Materials for Fiber Drawing

**DOI:** 10.3390/ma15031186

**Published:** 2022-02-04

**Authors:** Tanel Tätte, Medhat Hussainov, Mahsa Amiri, Alexander Vanetsev, Madis Paalo, Irina Hussainova

**Affiliations:** 1Institute of Physics, University of Tartu, Riia 142, 51014 Tartu, Estonia; mahsa.amiri@ut.ee (M.A.); alexander.vanetsev@ut.ee (A.V.); madis.paalo@fi.tartu.ee (M.P.); 2School of Engineering, Department of Mechanical and Industrial Engineering, Tallinn University of Technology Ehitajate tee 5, 19086 Tallinn, Estonia; medhat.hussainov@taltech.ee (M.H.); irina.hussainova@ttu.ee (I.H.)

**Keywords:** CNT doped composites, titanium oxide, sol-gel, rheology, elongational viscosity

## Abstract

A strategy of doping by multi-walled carbon nanotubes (MWCNT) to enhance mechanical strength and the electrical conductivity of ceramic fibers has nowadays attracted a great deal of attention for a wide variety of industrial applications. This study focuses on the effect of MWCNTs on rheological properties of metal alkoxide precursors used for the preparation of nanoceramic metal oxide fibers. The rheological behavior of MWCNT-loaded titanium alkoxide sol precursors has been evaluated via an extensional rheometry method. A substantial decrease in elongational viscosity and relaxation time has been observed upon an introduction of MWCNTs even of low concentrations (less than 0.1 wt.%). A high quality MWCNT/nanoceramic TiO_2_ composite fibers drawn from the specified precursors has been validated. The MWCNT percolation, which is mandatory for electrical conductivity (50 S/m), has been achieved at 1 wt.% MWCNT doping.

## 1. Introduction

Carbon nanotubes (CNT) of superior mechanical, electrical, and thermal properties [1,2,3,4] are frequently used in order to enhance the properties of a wide variety of materials, for example, ceramics and/or ceramic matrix composites (CMCs). The CNT added composites combine synergistic effects of their components. The addition of CNTs to the brittle Al_2_O_3_ enhances its toughness significantly, preserving a high-temperature stability and exceptional creep resistance [5]. Similarly, electrical conductivity of thermally and chemically highly stable but inherently dielectric TiO_2_ can be boosted enough for application of TiO_2_ as long-cycle-life anodes in the lithium-ion batteries [6,7].

One of the key challenges in the preparation of the reinforced composites is the uniform dispersion of CNTs in the matrix. Agglomeration and non-homogeneous distribution of nanotubes act as defects promoting stress-concentration and failure of materials [8]. Numerous techniques have been proposed to fabricate CNT-reinforced ceramic composites including chemical vapor decomposition (CVD) [9], plasma sintering [10], colloidal processing [11], electrophoretic deposition [12], sol-gel [13], etc. In principle, the methods can be classified as top-down and bottom-up approaches. The first approach is based on applying shear forces to distribute CNTs throughout preliminarily prepared matrix precursors, ultrasonic probes, ball milling, etc. In case of the bottom-up approach, for example, the sol-gel method, two constituents are mixed at the molecular level [14,15]. Solid composites with a desired structure are obtained as a result of gelation of an initial sol matrix, which is typically prepared from metal alkoxides [16]. From the point of view of the final product homogeneity, the bottom-up approaches are more anticipated as they allow homogenous dispersion of CNTs and better adhesion between CNTs and precursor particles. Besides, a sol-gel procedure offers a simple method for giving a final shape for the materials at a room temperature. During the sol-gel transition, the viscosity of sol-material increases until the material transforms elastic gels as a result of formation of a 3D gel network over the whole volume [17,18].

Previously we have demonstrated that chemical reactions of alkoxides of different metals proceed with significantly different kinetics [19]. For example, gelation of titanium butoxide sols occurs much more homogeneously with the formation of Van der Waals bonds in the beginning of process as compared to zirconium butoxide sols, which immediately lose a majority of alkoxy groups in a contact with water, therefore releasing a significant amount of alkoxy groups and forming strong covalent bonds between the metal-oxo-alkoxy sol particles. It also was demonstrated that, due to the large quantity of released alcohol, sol particles are free to rearrange and self-form into high quality ZrO_2_ gel microtubes with optical wave guiding properties [20].

Viscosity and rheological properties of sols and forming gels depend on the structure and can therefore be used to monitor the formation process and condition of the final gel materials [21,22]. Rheological study of sol materials is of a great importance for the controllable preparation of composites [23,24,25,26] In the wake of this research direction, it was found that sols and dispersions of nano objects in the liquid phase can change rheological characteristics of the latter in rather different ways depending on the chemical nature and geometry of both the liquid and the nanophase. Hammat et al. [27] have obtained Newtonian nanofluids with 2 wt.% concentration of Mg(OH)_2_ nanoparticles dispersed in ethylene glycol. The addition of 2 wt.% nanoparticles has led to a 60% increase in viscosity. On the other hand, Tuteja et al. [28] demonstrated that the addition of polystyrene nanoparticles to linear polystyrene leads to viscosity reduction, but only when the polymer molecules are entangling and confined. The authors confirmed that viscosity reduction can be attributed to an increase in the free volume. Chen et al. [29] observed an increase in matrix chain length N as a result of loading single-chain nanoparticles (SCNP) into the polystyrene, which amplifies the viscosity reduction effect. They have confirmed that the N-dependent viscosity reduction is associated with friction reduction, and the system with longer polymer chains will have a larger viscosity reduction. A theoretical model was also proposed to justify the viscosity ratio between the polymer/SCNP composite system, with a small SCNP loading, and the pure polymer system measured in an experiment. Changes in rheological properties are not restricted to the two mentioned patterns. An intermediate trend has also been reported for some nanofluids. Motahar et al. [30], have studied the rheological behavior of a system containing the TiO_2_ nanoparticles dispersed in n-octadecane. They have observed a Newtonian behavior for concentrations lower than 2 wt.%. By increasing the mass fraction of TiO_2_ nanoparticles to higher than 2 wt.%, the rheological data demonstrate a transition from Newtonian to non-Newtonian behavior.

These earlier examples prove that loading CNTs as nano objects into the liquid can significantly change its rheological properties according to CNT characteristics such as orientation, dimension, aspect ratio, concentration, and surface chemistry [31,32] as well as liquid ones. Hemmat et al. [33] studied the effect of hybrid nanofluids on the rheological behavior of the nanofluid containing multiwall carbon nanotube (MWCNT) and TiO_2_ nanoparticles and pointed out the significant importance of CNT addition on the rheological properties of the fluid. The nanofluids containing MWCNTs at the solid volume fraction of 0.75% exhibited an 100% increase in viscosity. Transition from Newtonian to non-Newtonian pattern as a result of increasing the number of loaded particles was also reported. A larger amount of carbon nanotubes in the nanofluid leads to more pronounced deviation from Newtonian behavior of the nanofluid. Similarly, in another study, Tian et al. [34] incorporated nanoparticles of CuO/MWCNTs into the base fluid of mixed water-ethylene glycol (EG) in order to study the changes in rheological behavior. They have reported that shortly after the addition of the nanoparticles, the fluid still showed a Newtonian behavior with an increase in viscosity. However, further addition of nanoparticles led to a transition in rheological behavior to the non-Newtonian pattern. Increasing the amount of loaded nanoparticles would have a more intense effect on the rheology of the nanofluid, which has been confirmed by Yan et al. [35]. They studied the rheological properties of MWCNTs–ZnO/water–EG system and reported that, increasing the volume fraction of loaded particles, the effect of added particles is intensified, and the non-Newtonian property is more likely to appear. Changes in the rheological behavior of CNT-loaded fluid can be attributed to the percolation of CNT aggregates. In polymer and hydrogel systems containing fillers such as nanoparticles and CNT, the minimum filler concentration that establishes a conductive path of connected particles is considered as percolation threshold [36,37]. These connected particles can affect various properties of the system including thermal and electrical conductivity as well as the rheological properties. Azizi et al. [38] have incorporated MWCNT into a system of polypropylene/polylactic acid (PP/PLA) and studied the changes in electrical and rheological properties after loading various amounts of MWCNT. With the amount of MWCNT reaching 2 wt.% (percolation threshold), the free charge carrier concentration grows dramatically and leads to a significant increase in electrical conductivity. Besides, Nadiv et al. [39] have investigated the integration of CNTs into an epoxy matrix and the changes in rheological properties as a result of forming a percolating network of CNT aggregates. At the percolation threshold, the amount of CNT aggregates exceeds the amount of the individual CNTs, so the aggregate-aggregate interaction hampers their bulk motion as well as the motion of the epoxy molecules within and around them. Therefore, the diffusive motion of epoxy molecules is strongly decreased in the close vicinity of the aggregates, and even at larger distances, leading to an increase in viscosity.

Not only the rheological properties of the sol and subsequent gel but also the mechanical properties and the electrical conductivity of the final materials, e.g., fibers prepared from these sols vary as a result of the increasing CNT load. Research findings confirm that the addition of CNTs into polymeric fibers increases their electrical and thermal conductivities [40]. In fact, the enhanced electrical conductivity of the composite fibers is also a result of the percolation of CNT network. Apart from CNT concentration, their orientation is another important parameter affecting percolation and, in turn, composite fiber electrical conductivity. Drawing fibers can change the orientation and, therefore, the percolation of the fillers [41]. Lu et al. [42] had observed that the electrical conductivity of CNT/polymeric fibers could be changed by their stretching, for example, experiments with PAN/CNT (4.4 wt.%) fibers had shown that stretching enables to increase their electrical conductivity from 0.05 to 0.38 S/m. However, even further stretching has led to a drastic decrease in electrical conductivity of the samples as it leads to breaking contacts between CNTs and, therefore, an interruption of their percolation. The alignment of CNTs may increase or decrease their percolation and thus result in an increase or a decrease in the electrical conductivity of the composites [40].

The rheological behavior of widely used metal alkoxide sols during their gelation has been poorly studied and thus their influence on other properties of the sol-derived fibers remains obscure. The aims of the current study is evaluation of the rheological behavior of MWCNT-loaded titanium alkoxide sols in order to use them as precursors in the preparation of MWCNT/nanoceramic TiO_2_ composite micro scale fibers. TiO_2_ is chosen as one of the most used ceramics due to its excellent mechanical and chemical stability, light weight, and promising photovoltaic and catalytic properties [43]. As discussed, the solventless titanium alkoxide fiber precursors are a suitable object for rheological measurements as they undergo gelation very homogeneously, with just a small loss in volume during the gelation. However, for many applications, e.g., electrodes in fuel cells or wear resistant materials, the properties of TiO_2_ composite materials should still be improved [44].We performed a systematic study of the rheological characteristics of MWCNT/TiO_2_ precursors using an extensional rheometry method developed for highly reactive materials. We aimed to improve electrical conductivity of the fibers drawn from the precursor, making them suitable for high technology applications such as optical devices, energy storage, etc.

## 2. Materials and Methods

### 2.1. Preparation of Precursors

The precursors were synthesized from titanium(IV)propoxide (Ti(OPr)_4_) (Sigma-Aldrich, St. Louis, MO, USA) dissolved in propanol at a room temperature. The reagent-grade propanol (Sigma-Aldrich Comp) was dried with CaH_2_ and distilled prior to use. The MWCNTs prepared by a catalyzed-chemical-vapor-deposition (CVD) with 10–20 nm diameter and 10–30 µm length were used as purchased from CheapTubes Inc, Cambridgeport, Cambridge, MA, USA.

To initiate the formation of titanium-oxo nanoparticles, propanol, containing 8% of water, was dropwise added to 10 g of Ti(OPr)_4_ to achieve the molar ratio of R(H_2_O/Ti(OPr)_4_) = 0.5. Subsequently, MWCNTs (e.g., 0.01 g) were added into the solution as a 10mL of visually homogeneous solution in dry propanol. Prior to the addition, the nanotube solution was ultrasonicated for 30 min inside a metric cylinder (e.g., 15 mL) by using a Hielscher UP200S (24 kHz and 200 W, Hamm, Germany) apparatus. After mixing, the excess of the solvent was removed from the suspension using a Büchi Rotavapor^®^ R210 (Landsmeer, Netherlands) equipped with a Büchi V700 membrane pump under 65 °C water bath and 1–5 torr vacuum and an ultrasonic bath (Elmasonic S 30/(H), Singen, Germany, 37kHz and 80 W) was used instead of ordinary Büchi heating bath. After removal of the propanol, the visually homogeneous samples with the six different CNT concentrations (0; 0.01; 0.02; 0.05; 0.1 and 1 wt.%) were obtained for further investigation.

### 2.2. Rheological Measurements

A developed technique for extensional rheometry [44] has been used in the present work. The main advantage of this technique is the possibility to carry out the rheological measurements in the reactor where the high moisture-sensitive fluids were synthesized. The liquid precursor thread was drawn directly in a 100 mL glass-made reaction bulb. Filament thinning under the combined actions of surface tension, gravity, and rheological effects was monitored through the bulb wall by a high-speed camera (MC1310 Mikrotron GmbH, Unterschleissheim, Germany) with maximum resolution 1280 × 1024 pixel at 502 frames per second in the mid-filament region. After the experiment, the obtained video frames were processed using a specially-developed software. As a result, the transient filament profile at a mid-filament area and the evolution of the minimum filament diameter until thread breakup were obtained.

### 2.3. Fibers Preparation and Characterization

After the rheological measurements, the fibers were drawn from the precursor samples by immersing a glass rod, ~5 mm in diameter, into it and pulling the rod off from the bulb at ~1 m/s speed [45]. The surface of the formed liquid precursor threads solidifies immediately in the lab atmosphere (relative humidity 20–50%, temperature 20 °C) after the reaction with water vapors in a surrounding atmosphere. This is accompanied by the formation of the bridging –O– bonds between metal-oxo nanoparticles. The prepared fibers were aged then for 24 h in atmospheric conditions. Multistep thermal treatment up to 200, 300, 400, 500, and 600 °C was performed to burn out organics and to densify the samples, keeping the samples for 5 h at each of temperature. Finally, dense fiber samples of 10–50 mm length and 5 to 70 µm diameter were obtained.

The transparency and morphology of the annealed fibers were evaluated via an optical microscope and a high-resolution scanning electron microscope (Helios NanoLab 600) equipped with a focused ion beam (FIB). Cutting the fibers along their axes via FIB was necessary to expose aligned CNTs inside the material. The electrical properties of the materials were measured by the 4-point method. The electrical contacts, made of In-Ga alloy, were used between the measuring probes and the fiber—the droplets of alloy, 0.5 mm in diameter, were placed on parallel gold-coated measuring probes with 1 mm spacing. The fibers for the measurements were placed perpendicularly across the probes, immersed into the droplets. The resistance was measured using a Keithley, Gorinchem source meter model 6517A and a Keithley voltmeter model 2400. The fiber diameters, needed in calculations of absolute values of their electrical conductivity, were picked from the obtained SEM images.

## 3. Results and Discussions

### 3.1. Preparation of Precursors

A detailed description of the synthesis of the metal-oxo-propoxide ceramic fiber precursors is detailed elsewhere [46]. Metal alkoxide precursors are highly reactive species, prone to react with water without catalysts. In fact, the reaction results in one-step hydrolysis-condensation transformation [47] associated with the profound restructuring of the molecules, as in the case described below [48,49], for example:3Ti_2_(OiPr)_8_ + 2H_2_O = 2Ti_3_O(OiPr)_10_ + 4iPrOH

The hydrolysis of metal alkoxide solutions in (humid) air, without any catalyst, has been denoted as “natural” hydrolysis. It has been shown in many works that hydrolyzed metal alkoxides exist as nanoscopic core-shell structures: sub-crystalline metal-oxo cores (up to 1–2 nm in size), stabilized by an alkoxy layer on their surface. The explanation for the formation of metal-oxo particles is proposed by V. Kessler et al. as micelles templated by self-assembly of ligand (MTSAL) [47]. We have demonstrated earlier that the obtained liquids exhibit non-Newtonian visco-elastic behavior. That behavior, required for their use as fiber drawing dopes, is possible due to the arrangement of primary nanoclusters into secondary elongated units, which undergo sliding when external forces are applied [45].

In this study, the method for the preparation of the fiber precursors was modified in order to incorporate CNTs into the final material. The preliminary 30 min ultrasonic agitation in a relatively small beaker using the Hielscher UP200S apparatus was necessary to break CNT bundles and agglomerates. Relatively mild sonication conditions (80 W) were applied during the solvent removal process. Solvent removal at the same, time with the help of a Buchi roratary evaporator, leads to increased viscosity. At the end of process, the achieved high viscosity 10–10,000 P [20] of drawable fiber precursors prevents MWCNTs from re-aggregating into bundles.

### 3.2. The Rheological Properties of the CNT-Doped Metal Alkoxide Precursors

For the calculation of the rheological parameters from the evolution of the minimum filament diameter, two types of approximations of the experimental data (Figure 1) were applied:
(i)approach for initial and intermediate stages of evolution described by the equation [50]:
Rmin = A × exp(−B × t) − C × t + D(1)(ii)approach at the final stage of the filament evolution described by linear law:
R_min_ = A × t + B(2)
where *t* is time and *A*, *B*, *C*, *D* are fitting parameters.

Figure 1 gives evidence that Equation (1) approximates experimental data very well only at initial and intermediate times reflecting the visco-elastic regime of filament evolution. Therefore, Equation (1) was used for calculation of the steady-state extensional viscosity *η_Eµ_* and longest relaxation time *λ_E_* of liquid:η_Eµ_ = σ/C(3)
λ_E_ = 1/3B(4)
where *σ* is the surface tension.

At the final stage of filament evolution, just before the break-up, the inner structure of the liquid becomes aligned and elongated. At this stage, the filament diameter varies in time similar to the variation of size known for a Newtonian liquid, i.e., linearly. The approximation of this stage by Equation (2) allows obtaining zero-shear viscosity η_s_ of a liquid with a specific inner structure such as aligned CNTs and precursor particles:η_s_ = −α × σ/A(5)
where the constant α = 0.0709.

The measurements were carried out at 15 mm rod elevation, which defined the length of the filaments. The additional tests were conducted at a 10 mm rod elevation for a pure sol precursor as well as for the sol precursors with CNTs concentration of 0.05 and 0.1 wt.% because of too short break-up times of filaments drawn from these samples. It was not possible to get data consistent with the behavior of the sample containing 1 wt.% CNTs because of the inhomogeneous nature of the material. The inhomogeneity leads to the quick collapse of the drawn, irregularly-shaped liquid threads before the rod reached the final position needed in measurements. The inhomogeneity could be explained by the bundles and ropes of CNTs, which remain in the material. Alternative explanations for the behavior could be related to the percolation of carbon nanotubes, which critically influence the flowing properties of the precursor at that high CNT concentration. 

All the measurements were run at least three times per test condition. Inaccuracy of rheological parameters calculations were no more than 6% for extensional viscosity, 10% for zero-shear viscosity, and 30% for the relaxation time.

Figure 2 shows an unexpected result in the experimental study of the precursors’ elongational properties. It can be seen that dispersion of a small amount of CNTs in precursors has led to a dramatic decrease in filament break-up time. The decrease in viscosity and relaxation time for the suspensions containing nanoparticles usually occur if the filler particles are smaller than the radius of gyration. However, in our case, the particle size (the CNTs were 10–20 nm in diameter) is far above the required 2 nm [46]. Such an effect of CNT’s concentration on the rheological parameters could be explained by fluid layers on the surface of dopant particles [51]. The chemical composition of the precursors and their weak interaction with CNTs leads to a reduced viscosity in the fluid layers on the particle surface and therefore decreases bulk viscosity. Viscosity reduction as a response to the addition of high aspect ratio solid additives with lowered friction on their surface is in good correlation with earlier results that have shown lowered viscosity of systems containing elongated particles, as reported by Chen et al. [29]. When outer force is applied on such system then the mass of the liquid matrix starts sliding on the surface of elongated filler particles. This results in a decrease of the bulk viscosity of the matter. On the basis of the obtained result, CNTs could be proposed as additives to modify the viscosity of high-viscosity fluids, for example, mineral oils, in order to lower their viscosity.

It has also been reported earlier that such kind of dependence in the filament break-up time on CNTs concentration in CNT blended epoxy resins could be explained by the formation of CNT clumps-aggregates, ropes, and bundles [52]. However, we have shown that the surface of the produced filaments was smooth and not affected by the clumps at all the concentrations of CNTs (Figure 3), except a heavily loaded sample containing 1 wt.% of CNTs. The evolution of the minimal filament diameter in time decreases gradually without significant heterogeneities (Figure 4). This indicates the fact that the shorter time of the filament break-up is not related to the non-homogeneous dispersion of CNTs.

The observed effect may be related to the addition of elongated objects such as single MWCNT and their ordered small bundles. To prove this hypothesis, we have studied the influence of CNTs concentration on the rheological parameters of the precursors (Figure 5). It is worth mentioning that the values of zero shear viscosity correspond to the shear viscosity of a liquid with preliminary aligned CNT and particles of the precursor, so that the values were obtained from an approximation of the experimental data with the help of Equation (1). The extensional and zero shear viscosities have the same tendency: a little increase in the case of concentrations equals to 0.01 wt.%, and a strong decrease (by several times) in the concentration of 0.05 wt.%. For larger concentrations (0.1 wt.%) of CNTs, the parameters do not change significantly. The relaxation time decreases with the increase in CNTs content.

### 3.3. CNT-Metal Oxide Fiber Preparation and Characterization

It was possible to draw lengthy fibers with a circular cross-section from all the precursors except the sample with the highest (1 wt.%) CNT concentration—the sample which also “failed” in rheological characterization. However, it was still possible to prepare some short filaments with a rough surface from that precursor—long enough for further FIB-SEM and conductivity analysis.

The heat treatment of the as-prepared materials was carried out after their aging in order to determine the temperature required to burn-off organics from oxide material while preserving carbon nanotubes. The thermal behavior of sol-gel prepared fibers is discussed in our earlier study [19], while a comparison of thermal oxidation of amorphous carbon vs. CNTs is discussed in [53]. It was shown also in this work that when using those particular kinds of CNTs purchased from Cheap Tubes Inc. (in the literature, there is significant evidence that the properties of the carbon nanotubes that are provided by different producers vary largely), thermal treatment at 350 °C in the air can be applied for that purpose. Application of the rather high temperature was possible because of the large diameter of used CNTs such as SWCNTs. As the evaluation of the distribution of CNTs in fiber material with optical microscopy was disturbed heavily by the black color and small diameter of the fibers, SEM imaging was used as the main tool for structural characterization. SEM images revealed that the surface of the fibers was generally smooth, although individual irregularities on the surface could be seen on fibers with CNTs concentrations of 0.1 wt.%. However, it was possible to find smooth areas even on the surface of filaments containing 1 wt.% of CNTs. As the number of surface irregularities was in correlation to the concentration of CNTs, it can be assumed that irregularities on the surface are most probably caused by CNT ropes and bundles that are still present in the matrix. SEM imaging of the area, cut using FIB, proves clearly that most of the tubes in the material are dispersed as individual tubes, small ropes, or very small bundles (Figure 6).

When 1 wt.% of CNTs was inserted into the titanium dioxide matrix, the CNT percolation, sufficient for electrical conductivity of the composite material, was detected. The conductivity of the samples containing 1 wt.% of CNTs, heat-treated up to 400 °C, was in the range of 50 S/m. The latter was at least five orders in magnitude higher value than that of fibers with lower CNT concentration. When the fibers, containing 1 mass% of CNTs, were heat-treated up to 600 °C then the conductivity decreased to 10–6 S/m. Visual observation of the samples under transmitting optical microscopy showed transformation of their color from black (the samples heat-treated at 500 °C) to opaque white (heat-treated at 600 °C) at the same. Transition of color could be explained by oxidation of CNTs in the content. It has been shown earlier that CNTs decompose at thermal treatment up to 375–450 °C. Exact decomposition temperature depends on their crystal structure, number of walls and defects. The fact that the samples, heat-treated up to 500 °C, were still black in color demonstrates that TiO_2_ matrix protects incorporated CNTs against oxygen. The CNT percolation threshold between 0.1 and 1 wt.% indicates that the used CNT dispersion method was highly effective. For comparison, the CNT network formation in polymer-based composites usually takes place when CNT concentration is in range from 0.5 to 10 wt.% [54]. The emergence of a CNT percolation network as the response to the increased CNTs concentration can be followed from SEM images of FIB-cut areas of fibers presented in Figure 6. Below 0.05 wt.%, the CNTs remained scattered inside the matrix, while in the fibers reinforced with 0.1 wt.% CNTs, individual areas of percolating networks could be seen.

It can be seen from Figure 6b,c that, during the pulling process, CNTs are oriented along the direction of the fiber axis by elongational forces. In addition, it seems from the SEM-FIB analysis that big “loose” bundles are stretched out in the pulling direction. The elongation of particles also demonstrates that highly viscous matrix material solidifying via the formation of bridging –O– bonds between metal-oxo nanoparticles prohibits to lose of orientation of CNTs. Although in the current study mechanical properties of the composites were not measured, it is known from the experiments with polymer/CNT fibers that the orientation of the carbon nanotubes enhances the overall mechanical strength of the material [55].

## 4. Conclusions

Ti-oxo-propoxide precursors were prepared from Ti-propoxide as a result of the reaction with water. CNTs were ultrasonicated in propanol and added to the precursors in order to obtain suitable materials for the fabrication of CNT-TiO_2_ composite fibers. The fibers were drawn directly from the precursors, aged, and thermally treated. As one of the most important parameters of the fiber precursor material, the viscoelastic properties of the precursors were studied prior to fiber processing. Rheological studies carried out by applying unique elongational rheometry—a novel set-up proposed earlier—demonstrated a decrease in elongational viscosity and relaxation time with an increase in CNT content at low concentrations (less than 0.1 wt.%). The zero-shear viscosity of the pre-aligned CNT-doped precursors with increasing CNTs content was also decreased. Careful heat-treatment of the materials at 350 °C resulted in the oxidation of organic additives and amorphous carbon from the final CNT-doped TiO_2_ ceramic materials. The percolation of CNTs for good electrical conductivity, 50 S/m, was achieved by adding 1 wt.% of CNTs to the precursor materials. The percolation-based conductivity mechanism was supported by the fact that heat-treatment up to 500 °C caused six orders of magnitude drop-down in conductivity down to as low as 10–6 S/m as a result of the oxidation of the carbon nanotube network.

## Figures and Tables

**Figure 1 materials-15-01186-f001:**
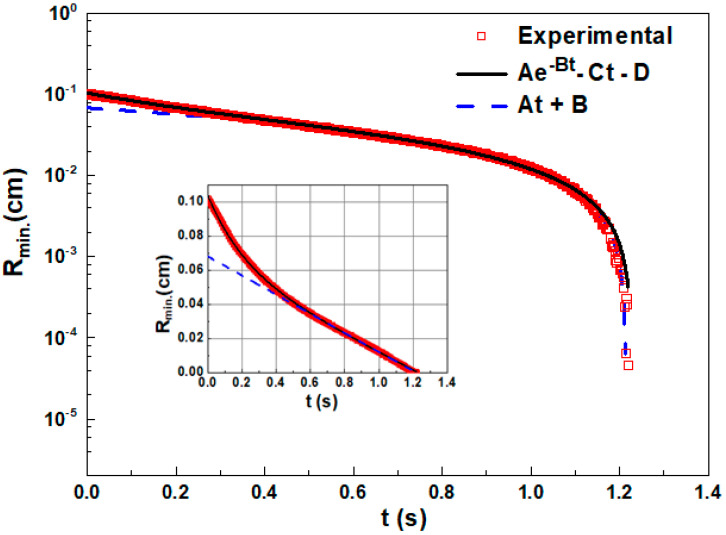
Evolution of the minimal filament diameter with time for CNT-free precursor at the rod elevation of 15 mm. Lines show the fitting of the experimental data by two equations: R_min_ = A × exp(−B × t) − C × t + D and R_min_ = A × t + B. Insertion is in the linear axis for the same data.

**Figure 2 materials-15-01186-f002:**
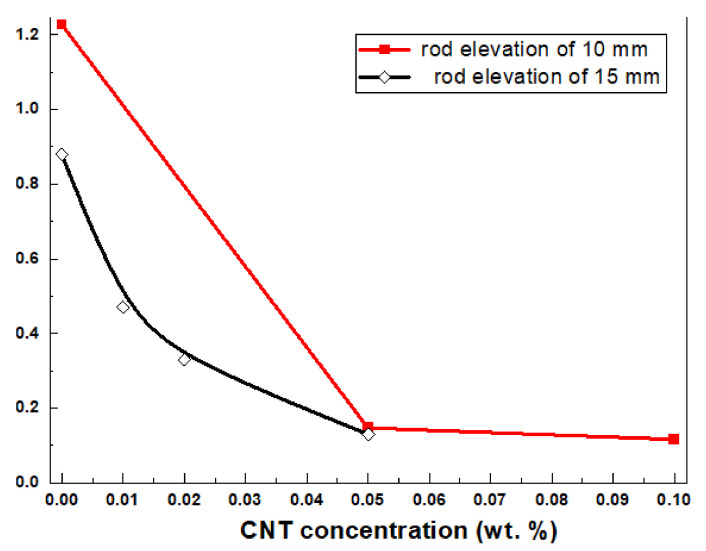
Effect of concentration of CNTs on the filament break-up time t_b_.

**Figure 3 materials-15-01186-f003:**
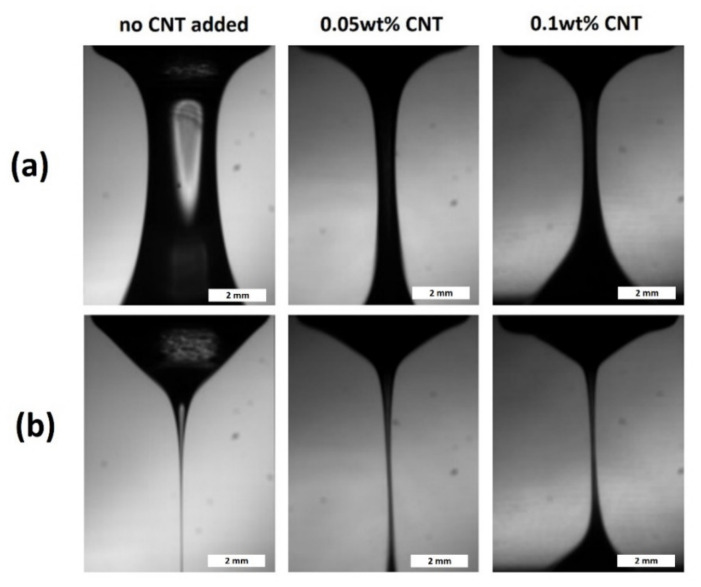
The shapes of the filaments for the liquids without CNT, 0.05 wt.% CNT, and 0.1 wt.% CNT (the left, middle and right column, respectively) at 2 times: (**a**) at 20 ms after rod cessation; (**b**) at 20 ms before filament break-up. The lifetimes of the filaments (t_b_) were 1.23 s, 0.16 s, and 0.1 s, respectively. The rod was elevated for 10 mm.

**Figure 4 materials-15-01186-f004:**
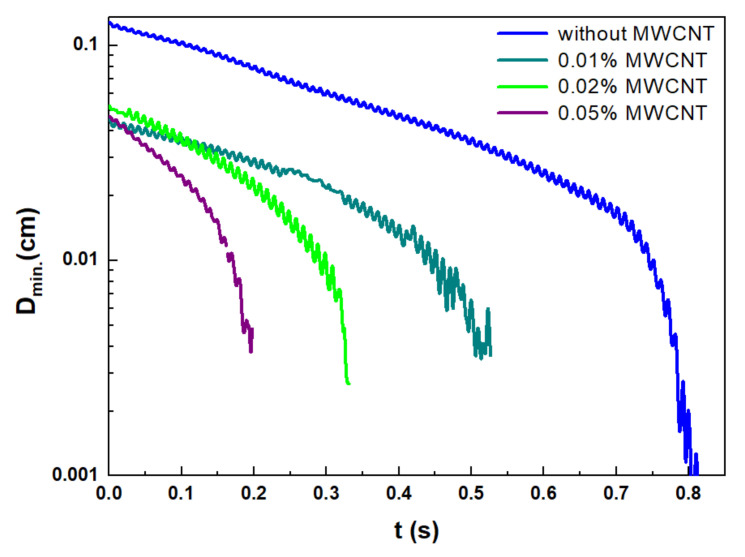
The evolution of the minimal diameter of the pure precursor and precursors with 3 different CNT concentrations: 0.01, 0.02, 0.05 wt.%. The rod elevation was 15 mm.

**Figure 5 materials-15-01186-f005:**
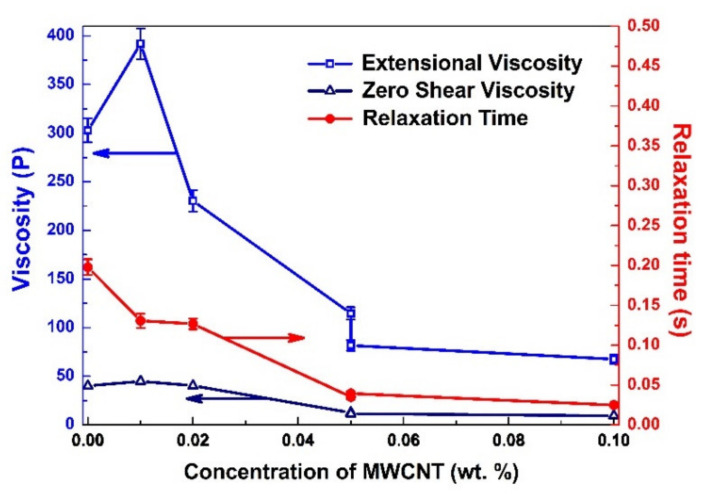
The rheological parameters (steady-state extensional viscosity *η_Eµ_*, zero shear viscosity *η_s_*, and longest relaxation time *λ_E_*) as a function of CNT concentration.

**Figure 6 materials-15-01186-f006:**
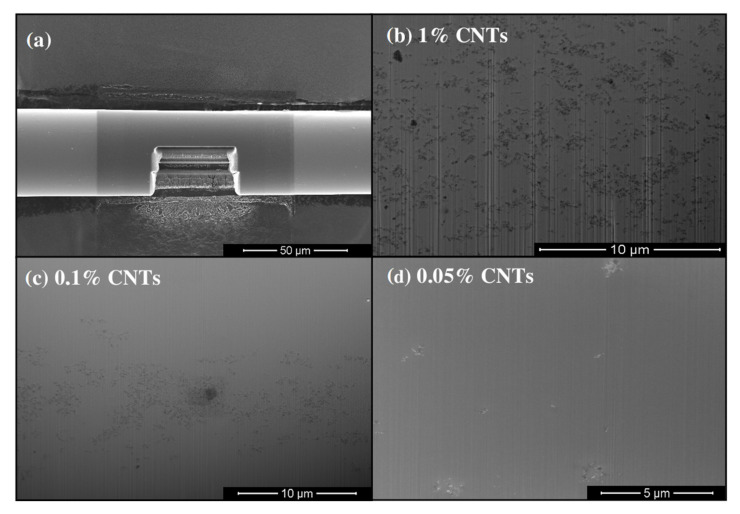
SEM images of (**a**) a fiber cut by FIB for monitoring of CNTs distribution inside the material; (**b**–**d**) images of materials with different MWCNTs concentration.

## Data Availability

Some or all data, models, or code that support the findings of this study are available from the corresponding author upon reasonable request.

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
