# Peer review of "Rheological Properties of MWCNT-Doped Titanium-Oxo-Alkoxide Gel Materials for Fiber Drawing"

_materials, 2022, doi:10.3390/ma15031186_

Round 1

Reviewer 1 Report

Dear Authors,

The work entitled "Rheological properties of MWCNT-doped titanium-oxo-alkoxide gel materials for fiber drawing" by Tanel Tätte, Medhat Hussainov, Mahsa Amiri, Alexander Vanetsev, Madis Paalo and Irina Hussainova refers to the influence of multi-walled carbon nanotubes (MWCNTs) on the rheological properties of metal alkoxide precursors, used in the preparation of nanoceramic metal oxide fibers.

It should be stated that the content of the presented work corresponds to the topic defined in its title. The manuscript has the right structure and is written correctly, it deals with current issues and can be useful for people dealing with the above-mentioned issues. The proposed methodology is correct and leads to the solution to the problem presented in the title. Although the work has an appropriate number of literature references, there is no properly conducted discussion of the obtained results.

The authors of the work focused primarily on describing the obtained results, but there is almost no immediate comparison with other results of other authors or a short discussion on this topic - this should be supplemented.

Figure 1 should be located closer to its quotation in the text, it will also improve the readability of the text, because the caption under the Figure is on a different page than the figure itself.

In addition, the manuscript is not written in accordance with the journal's requirements, e.g.:

  1. When specifying the name of the enterprise, the city of its seat and the country should be provided in brackets.
  2. At the end of the manuscript, please indicate, first of all, Author Contributions and refer to the Institutional Review Board Statement, Informed Consent Statement and Data Availability Statement.
  3. All literature items are incorrectly written, they should be corrected as required.

The manuscript preparation requirements available on the journal's website should be read in detail and applied carefully.

Before publishing the manuscript, I recommend to remove the incorrect issues mentioned above.

Best regards,

Author Response

Dear Reviewer,

Thank you very much for your comments. Taking into account all of their suggestion we improved our manuscript as much as it was possible in that quite short time. Thus please find our answers to your concreate questions and suggestions.

We made some small changes, mostly into introduction, also on our own initiative. These definitely will not change the content of story but makes it more easy to read. 

Comments:

The authors of the work focused primarily on describing the obtained results, but there is almost no immediate comparison with other results of other authors or a short discussion on this topic - this should be supplemented.

Figure 1 should be located closer to its quotation in the text, it will also improve the readability of the text, because the caption under the Figure is on a different page than the figure itself.

Thanks for the note, we shifted the figure a bit in the text.

When specifying the name of the enterprise, the city of its seat and the country should be provided in brackets.- At the end of the manuscript, please indicate, first of all, Author Contributions and refer to the Institutional Review Board Statement, Informed Consent Statement and Data Availability Statement.

Thanks for the note, we have applied this change.

All literature items are incorrectly written; they should be corrected as required.

Thanks for the note, we have applied this change.

Kind regards

Reviewer 2 Report

The manuscript by Tätte and co-workers investigates the rheological behavior of MWCNT-loaded titanium alkoxide sol precursors using extensional rheometry towards processing of MWCNT/nanoceramics TiO2 composites into micro scale fibers. The key highlights are

  • decrease in elongational viscosity, relaxation time and zero-shear viscosity upon increase in MWCNT concentrations (less than 0.1 wt %).
  • percolation of CNTs resulted in good electrical conductivity of ~ 50S/m with an addition of 1wt% of CNTs to the precursor materials.

Overall, this is contemporary research attempt in the context of CNT loaded nanocomposites. Although the paper is organized and well written, it lacks some important characterization details to support the claimed objectives and conclusions. This raise concern about whether the system being studied is well-suited for obtaining an incisive conclusion. I believe that the authors should address all the comments appropriately before making any further judgement on the manuscript. My comments are listed below.

Major

  1. a) In the introduction part the authors state that “We aimed to improve both optical properties and electrical conductivity of the fibers drawn from the precursor making them suitable for high technology applications like optical devices, energy storage, etc.” While the conductivity part is addressed in the conclusion to explore the material’s future potential in charge storage and transport, the scope of its use and study to potentially be employed in an optical device is missing both from the conclusion and main text. I am not sure what exactly is optical property improvement the authors are referring to for make them suitable to be explored in designing high tech optical devices.
  2. b) The sentence “High viscosity (10-10000P [16]) of drawable precursors prevents MWCNT from re-aggregating into bundles after ultrasonication.” is confusing. Is it the high viscosity or mechanical agitation (via ultrasonication) that prevents the reaggregation here?
  3. c) Please include the characterization details (TEM study) towards estimation of particle size of CNTs used for this work.
  4. d) The results in fig. 3 showing the dramatic decrease in filament break-up time upon dispersion of a small amount of CNTs is very interesting. Could the viscosity modification (as claimed by the authors) be carried out in a more controlled manner over a broad range of concentration. For example, what happened at an intermediate concentration such as at 0.03 wt% of CNT? The datapoints for the 0.03 wt.% CNT concentration is missing from fig. 2 and fig. 4.
  5. e) How are the images in fig. 3 obtained? Please address in the fig. caption and include the respective scale bars.
  6. f) Please provide the SEM images for other concentrations as well to understand the extent of percolation.
  7. g) Please include the electrical conductivity data for the 1wt% concentrations and include the details of the measurements. Why was it only measured for the 1 wt% sample when the authors claim to have more irregularities from the same? Also, what about the electrical conductivities for the other concentrations?

Minor

  1. a) Page 10- Please replace mass% with wt% to maintain consistency throughout the main text.
  2. b) Please provide appropriate references for the statement “In case of the bottom-up approach, for example, the sol-gel method, two constituents are mixed at the molecular level” in the introduction section. Direct evidence for molecular level mixing in sol-gel phenomena (specifically for obtaining thermally and mechanically improved conductive fibers) have been reported in ACS Appl. Mater. Interfaces 2016, 8, 39, 26176 and Chem. Eur. J. 2018, 24, 6217.
  3. c) Page 10 (line 360) – “When the fibers containing 1 mass% of CNTs were heat-treated up to 600 ºC, the resistance decreased down to 10-6 S/m”. It’s the conductivity that should decrease. Please correct the typo.
  4. d) Page 10 (line 360) – Figure 6 C and B should be addressed as Figure 6B and C.

Author Response

Dear Reviewer,

Thank you very much for your comments. Taking into account all of their suggestion we improved our manuscript as much as it was possible in that quite short time. Thus please find our answersyou’re your concreate questions and suggestions.

We made some small changes, mostly into introduction, also on our own initiative. These definitely will not change the content of story but makes it more easy to read. 

Comments:

  1. In the introduction part the authors state that “We aimed to improve both optical properties and electrical conductivity of the fibers drawn from the precursor making them suitable for high technology applications like optical devices, energy storage, etc.” While the conductivity part is addressed in the conclusion to explore the material’s future potential in charge storage and transport, the scope of its use and study to potentially be employed in an optical device is missing both from the conclusion and main text. I am not sure what exactly is optical property improvement the authors are referring to for make them suitable to be explored in designing high tech optical devices.

We would like to thank editor to note this. To tell the truth, we had several optical analysis in mind initially, but finally we focused on electrical properties of obtained materials. Thus, we removed this part from introduction at all. Still, we added visual estimation of transparency to the paper. It is written now into the experimental part:

Visual observation of the samples under transmitting optical microscopy shown that samples were black in color up to 500 ºC. Thermal treatment up to 600 ºC resulted in their transition white in color due to oxidation of carbon nanotubes. It has shown earlier that CNTs decompose at thermal treatment up to 375-450C. The fact that the samples, heat-treated up to 500 C were still black in color demonstrates that TiO2 matrix around them protects the tubes against oxygen.

  1. The sentence “High viscosity (10-10000P [16]) of drawable precursors prevents MWCNT from re-aggregating into bundles after ultrasonication.” is confusing. Is it the high viscosity or mechanical agitation (via ultrasonication) that prevents the reaggregation here?

Thank you to note this. We tried to make our message more clear, making changes to the chapter:

In this study, the method for the preparation of the fiber precursors was modified in order to incorporate CNTs into the final material. The preliminary 30 minutes ultra-sonic agitation in a relatively small beaker by using the Hielscher UP200S apparatus was necessary to break CNT bundles and agglomerates. Relatively mild sonication conditions (80W) were applied during the solvent removal process. Solvent removal at the same time with the help of Buchi rotary evaporator will lead to increased viscosity. At the end of the process, achieved high viscosity (10-10000P [16]) of drawable fiber precursors prevents MWCNTs from re-aggregating into bundles. 

  1. Please include the characterization details (TEM study) towards estimation of particle size of CNTs used for this work.

We initially discussed adding such an image, but we decided not to do it as TEM enables us to achieve very good resolution very locally in order to visualize a single tube or a small number of tubes. However, we found that TEM images are not that good for a general overview of materials.

It is also visible from added SEM image how the material is divided into bundles, ropes, and single tubes.

4.The results in fig. 3 showing the dramatic decrease in filament break-up time upon dispersion of a small amount of CNTs is very interesting. Could the viscosity modification (as claimed by the authors) be carried out in a more controlled manner over a broad range of concentration. For example, what happened at an intermediate concentration such as at 0.03 wt% of CNT? The data points for the 0.03 wt.% CNT concentration is missing from fig. 2 and fig. 4.

Yes, that influence of CNTs for viscosity was a truly serendipitous result. Unfortunately, it would be complicated to repeat all the measurements and calculations in that short time especially as one of our co-authors left the institute. However, this is a good note and we plan to take it into account it in our later studies. Our plans for the future include studies of the effect on carbon nanotubes with different surface treatments to see how surface energy influences the behavior of the matter.

  1. e) How are the images in fig. 3 obtained? Please address in the fig. caption and include the respective scale bars.

Thank you to note that we added scale bars for the images and also some more technical descriptions into the text.

  1. f) Please provide the SEM images for other concentrations as well to understand the extent of percolation.

We have clarified more about percolation, but it was not possible for us to apply more SEM images.

  1. g) Please include the electrical conductivity data for the 1wt% concentrations and include the details of the measurements. Why was it only measured for the 1 wt% sample when the authors claim to have more irregularities from the same? Also, what about the electrical conductivities for the other concentrations?

Thank you to ask that question. We clarified these aspects in the text.

Minor:

  1. Page 10- Please replace mass% with wt% to maintain consistency throughout the main text.

Thank you to note that we made these changes.

  1. Please provide appropriate references for the statement “In case of the bottom-up approach, for example, the sol-gel method, two constituents are mixed at the molecular level” in the introduction section. Direct evidence for molecular level mixing in sol-gel phenomena (specifically for obtaining thermally and mechanically improved conductive fibers) have been reported in ACS Appl. Mater. Interfaces 2016, 8, 39, 26176 and Chem. Eur. J. 2018, 24, 6217.

Thank you, we included the citations.

  1. Page 10 (line 360) – “When the fibers containing 1 mass% of CNTs were heat-treated up to 600 ºC, the resistance decreased down to 10-6 S/m”. It’s the conductivity that should decrease. Please correct the typo.

Thank you to note we made the change.

  1. Page 10 (line 360) – Figure 6 C and B should be addressed as Figure 6B and C.

Thank you to note we made the change.

Kind regards

Round 2

Reviewer 2 Report

I believe that the revised manuscript would be of interest of readership for the journal. I recommend acceptance of the manuscript for publication.